# BMI1-Inhibitor PTC596 in Combination with MCL1 Inhibitor S63845 or MEK Inhibitor Trametinib in the Treatment of Acute Leukemia

**DOI:** 10.3390/cancers13030581

**Published:** 2021-02-02

**Authors:** Katja Seipel, Basil Kopp, Ulrike Bacher, Thomas Pabst

**Affiliations:** 1Department for Biomedical Research (DBMR), University of Berne, 3008 Bern, Switzerland or katja.seipel@insel.ch (K.S.); basil.kopp@students.unibe.ch (B.K.); 2Department of Medical Oncology, University Hospital Berne, 3010 Bern, Switzerland; 3Department of Hematology, University Hospital Berne, 3010 Bern, Switzerland; veraulrike.bacher@insel.ch

**Keywords:** acute myeloid leukemia (AML), B-cell-specific Moloney murine leukemia virus integration site 1 (BMI1), hematopoietic progenitor cell antigen cluster of differentiation 34 (CD34), meningioma 1 (MN1), myelodysplastic syndrome (MDS), myeloid cell leukemia 1 (MCL1), tumor suppressor protein 53 (TP53)

## Abstract

**Simple Summary:**

Prognosis for acute myeloid leukemia (AML) patients is poor, particularly in *TP53* mutated AML, secondary, relapsed, and refractory AML, and in patients unfit for intensive treatment, thus highlighting an unmet need for novel therapeutic approaches. Targeting the stem cell oncoprotein BMI1 in leukemic cells may represent a promising novel treatment option for poor risk AML patients, especially in combination with other targeted therapies. Here we tested the BMI1 inhibitor PTC596 in combination with a variety of targeted therapies in AML cell lines and patient samples in vitro. In addition, we defined the biomarkers of response to the combination treatments in the leukemic cells. The combination treatment with the BMI1 inhibitor PTC596 and the MCL1 inhibitor S63845 may be an effective treatment in CD34+ adverse risk AML with elevated *MN1* gene expression and MCL1 protein levels, while combination treatment with BMI1 inhibitor PTC596 and the MEK inhibitor trametinib may be more effective in CD34+ adverse risk AML with elevated *BMI1* gene expression and MEK protein levels. The determination of gene and protein expression levels in leukemic cells as biomarkers of response to targeted combination therapies may be helpful to optimize treatment efficacy.

**Abstract:**

Purpose: Prognosis for acute myeloid leukemia (AML) patients is poor, particularly in *TP53* mutated AML, secondary, relapsed, and refractory AML, and in patients unfit for intensive treatment, thus highlighting an unmet need for novel therapeutic approaches. The combined use of compounds targeting the stem cell oncoprotein BMI1 and activating the tumor suppressor protein p53 may represent a promising novel treatment option for poor risk AML patients. Experimental Design: The BMI1 inhibitor PTC596, MCL1 inhibitor S63845, and MEK inhibitor trametinib, as well as the p53 activator APR-246 were assessed as single agents and in combination for their ability to induce apoptosis and cell death in leukemic cells. AML cells represented all major morphologic and molecular subtypes including *FLT3-ITD* and *FLT3* wild type, *NPM1* mutant and wild type, as well as *TP53* mutant and wild type AML cell lines and a variety of patient derived AML cells. Results: AML cell lines were variably susceptible to PTC596 and to combination treatments with PTC596 and MCL1 inhibitor S63845, MEK inhibitor trametinib, or *TP53* activator APR-246, independent of *TP53* mutational status. Susceptibility of patient samples for PTC596 in combination with S63845 or trametinib was significant for the majority of adverse risk primary and secondary AML with minimal efficacy in favorable risk AML, and correlated significantly with CD34 positivity of the samples. *BMI1* and *MN1* gene expression, and MCL1 and MEK1 protein levels were identified as biomarkers for response to PTC596 combination treatments. Conclusions: The combination of PTC596 and S63845 may be an effective treatment in CD34+ adverse risk AML with elevated *MN1* gene expression and MCL1 protein levels, while PTC596 and trametinib may be more effective in CD34+ adverse risk AML with elevated *BMI1* gene expression and MEK protein levels.

## 1. Introduction

Acute myeloid leukemia (AML) is a clonal blood malignancy characterized by arrested maturation and abnormal proliferation of hematopoietic precursor cells. At the cellular level, specific genetic and epigenetic alterations lead to changes in cellular signaling pathways, including the common inactivation of the *TP53* tumor suppressor axis, thereby contributing to the blockade of differentiation and accumulation of leukemic blasts in the blood and bone marrow [1].

The meningioma 1 (*MN1*) gene is expressed in CD34 positive hematopoietic stem cells and down-regulated during myeloid differentiation [2]. Increased *MN1* gene expression in myeloid stem cells leads to enhanced proliferation and loss of myeloid differentiation [3]. *MN1* overexpression has been linked to shorter overall (OS) and disease-free survival (DFS) of patients diagnosed with acute myeloid leukemia with normal cytogenetics [4,5,6,7,8]. Correlation analysis of *MN1* with myeloid gene expression levels revealed the positive association of *MN1* and *BMI1*, *CD34*, *FOXP1*, and *MDM2* expression indicating the stem cell gene BMI1 as putative MN1 target gene [9]. While there are currently no compounds targeting MN1, inhibitors targeting BMI1 are being tested in clinical studies for various solid tumors.

B-cell-specific Moloney murine leukemia virus integration site 1 (*BMI1*), the first functional gene in the PcG family, was identified in a mouse lymphoma in 1993. BMI1 induces self-renewal of hematopoietic (HSC) and leukemic stem (LSC) cells [10]. *BMI1* gene expression levels correlate well with progression and prognosis of myelodysplastic syndrome (MDS) and acute myeloid leukemia (AML) [11]. BMI1 plays an important role in the development of malignant tumors including solid tumors of the brain, breast, colon, head and neck, liver, lung, and prostate, as well as hematologic malignancies including lymphoma and leukemia [12,13]. Increased *BMI1* expression in tumor cells is associated with poor prognosis. BMI1 is a novel target for cancer therapy [14] and may be a valid target in AML therapy. BMI1 inhibition eliminates cancer stem cells [15]. PTC596 is a novel BMI1 inhibitor currently in clinical trials for ovarian, fallopian, and peritoneal cancer, glioma, and advanced solid tumors (NCT02404480, NCT03605550, NCT03761095). PTC596 is tolerable with manageable gastrointestinal side effects [16].

Targeting mutated *TP53* is a novel approach to restore the crucial p53 tumor suppressor function. APR-246 triggers an upregulation of genes involved in cell cycle control and apoptosis in both *TP53* mutant and wild type cancer cells [17]. APR-246 has various other effects, including induction of oxidative and ER stress [18]. The first-in-human study of APR-246 in hematologic malignancies (NCT00900614) demonstrated good tolerance and a favorable pharmacokinetic profile, with upregulation of p53 target genes and induced apoptosis [19]. Three clinical trials have been recruiting to test the safety and efficacy of APR-246 treatment in advanced esophageal carcinoma (NCT02999893), high grade ovarian cancer (NCT02098343), and mutant *TP53* hematologic myeloid malignant disease (NCT 03072043). Finally, a phase I/II study to investigate the safety and clinical activity of APR-246 in combination with a BRAF inhibitor in patients with mutant-BRAF unresectable metastatic melanoma resistant to anti-BRAF/anti-MEK inhibitors has started (NCT03391050).

In previous studies, we observed a considerable anti-leukemic efficacy of the MCL1-inhibitor S63845 and the MEK inhibitor trametinib. Hematological cells with susceptibility to the single compounds as well as to the combined treatment were defined by elevated MCL1- and MEK-protein levels, independent of the mutational status of *FLT3* and *TP53* [20].

Here we investigated the combined treatment of PTC-596 with the MCL1-inhibitor S63845 and the MEK-inhibitor trametinib, as well as the p53 activator APR-246 on AML cells in order to identify a potentially effective treatment specifically for aggressive AML, in particular in patients with refractory disease and in patients unfit for intensive chemotherapy. The study might provide the rationale for initiating a clinical study in aggressive AML evaluating PTC596 combination therapies.

## 2. Materials and Methods

### 2.1. Patient Samples

Mononuclear cells of AML patients diagnosed and treated at the University Hospital, Bern, Switzerland, between 2005 and 2018 were included in this study. Informed consent from all patients was obtained according to the Declaration of Helsinki, and the studies were approved by decisions of the local ethics committee of Bern, Switzerland, decision number #221/15. Peripheral blood mononuclear cells (PBMCs) and bone marrow mononuclear cells (BMMCs) were collected at the time of diagnosis before initiation of treatment. The AML cells were analyzed at the central hematology laboratory of the University Hospital Bern according to state of the art techniques [21]. Mutational screening for FLT3, NPM1, TP53, and conventional karyotype analysis of at least 20 metaphases were performed in all samples. The recent samples were analyzed by NGS sequencing of the myeloid panel genes.

### 2.2. AML Cell Lines

OCI-AML2 (AML-M4, FLT3wt, DNMT3A R882C, NPM1wt, TP53wt); OCI-AML3 (AML-M4, FLT3wt, DNMT3A R882C, NPM1mut, TP53wt), MOLM-13 (AML-M5, t(9;11), FLT3-ITD, TP53wt), MOLM-16 (AML-M0, FLT3wt, TP53mut), ML-2 (AML-M4, t(6;11), FLT3wt, TP53mut), PL-21 (AML-M3, FLT3-ITD), and HL-60 (AML-M2, FLT3wt, *TP53* null) cells were supplied by the Leibniz Institute DSMZ, German Collection of Micro-organisms and Cell Cultures. AML cells were grown in RPMI 1640 media (SIGMA-ALDRICH, St. Louis, MO USA) supplemented with 20% fetal bovine serum (FBS, Biochrom GmbH, Germany) in a standard cell culture incubator at 37 °C with 5% CO_2_.

### 2.3. Cytotoxicity Assays

For assays with AML cell lines, cells were plated at a density of 5 × 10^5^/mL and treated with targeted compounds or conventional induction therapy. For assays with patient derived mononuclear cells, the cells were cultured for 2 h prior to treatment. The BMI1 inhibitor PTC596, the FLT3 inhibitor midostaurin (PKC412), the MCL1 inhibitor S63845, the MEK inhibitor trametinib, and the p53 activator APR-246 were purchased at MedChemExpress (Monmouth Junction, NJ, USA). Conventional induction therapy consisted of equimolar solution of cytarabine and idarubicin purchased at Sigma-Aldrich (St.Louis, MO, USA) and SelleckChem (Houston, TX, USA). Cell viability was determined after 20 h of treatment using the MTT-based cell proliferation kit I (Roche). This time point was selected because the cellular responses were effectual for the calculation of combination indexes after 20 h of treatment with two compounds in leukemic cells. For AML cell lines, four independent assays (biological replicates) with four measurements (technical replicates) per dosage were performed. For hematological patient samples, two independent assays with three technical replicates per dosage were performed. For the calculation of combination indexes three dosages of PTC596 and two dosages of the other compounds were applied alone and in combination. Combination indexes were calculated on Compusyn software. Data are depicted as XY graphs, column plots, or scatter plots with mean values and SD. Statistical analysis was done on GraphPad Prism (GraphPad Software, San Diego, CA, USA) in grouped analysis and significance calculated by t-test for column graphs or Mann–Whitney test for scatter plots.

### 2.4. Measurement of mRNA Expression by qPCR

RNA was extracted from AML cells and quantified using qPCR. The RNA extraction kit was supplied by Macherey-Nagel, Düren, Germany. Reverse transcription was done with MMLV-RT (Promega, Madison, WI, USA). Real-time PCR was performed on the ABI7500 Real-Time PCR Instrument using ABI universal master mix (Applied Biosystems, Austin, TX, USA) and gene specific probes Hs01104728_m1 (ABL1), Hs00180411_m1 (BMI1), Hs00923894_m1 (CDKN2A), Hs00159202_m1 (MN1), and Hs02758991_g1 (GAPDH).

Measurements for BMI1 and MN1 were normalized with ABL1 values, measurements for CDKN2A were normalized with GAPDH values (ddCt relative quantitation). Assays were performed in three or more independent experiments. Statistical analysis was done on GraphPad Prism software using two-tailed t-tests. Data are depicted in column bar graphs plotting mean with SD values.

### 2.5. Measurement of Protein Levels by Enzyme-Linked Immunosorbent Assay (ELISA)

Protein extraction was done according to standard protocol. In short, cell pellets were lysed in RIPA buffer. GAPDH, MCL1, and MEK1 protein levels were determined with double-antibody Sandwich ELISA (SEB932Hu, SEC615Hu, SED559Hu, Cloud-Clone Corp., Houston, TX, USA). MCL1 and MEK1 values were normalized with GAPDH values. Two independent assays with three technical replicates were performed per sample. Statistical analysis was done using averaged normalized values and linear regression on GraphPad Prism software. Data are depicted as XY graphs with linear regression.

### 2.6. Antibodies and Cytometry

Staining for apoptosis was done using AnnexinV-CF488A (Biotium, Germany) in AnnexinV buffer and Hoechst 33,342 (10 μg/mL) for 15 min. at 37 °C, followed by several washes. Propidium iodide was added shortly before imaging on the Nucleocounter NC-3000 (ChemoMetec, Denmark). For cell cycle analysis, cells were incubated in lysis buffer with DAPI (10 μg/mL) for 5 min. at 37 °C and analyzed on NC-3000 imager.

## 3. Results

### 3.1. Susceptibility of AML Cell Lines to PTC596 and APR-246

To determine the sensitivity of AML cells to the BMI-1 inhibitor PTC596 and the p53 activator APR-246 AML cells were subjected to in vitro cytotoxicity assays. AML cells were treated with the compounds for 20 h in dose escalation experiments before cell viability assessment. The AML cell lines covered the major morphologic and molecular subtypes including, particularly, *FLT3-ITD* and *FLT3* wild type, *NPM1* mutant and wild type, as well as *TP53* wild type, mutant, hemizygous, and null cells (Table 1).

The susceptibility to PTC596 was elevated in *TP53* null HL-60 cells with IC50 of 220 nM, intermediate in *TP53* wild type MOLM-13 and OCI-AML3 cell lines with IC50 values of 300–500 nM, and rather low in *TP53* mutant cells lines PL-21, MOLM-16, and SKM-1 with IC50 values of 800–1200 nM. ML-2 cells with *MLL-AF6* transfusion oncogene and adverse risk were least susceptible to PTC596 with IC50 at 1500 nM (Figure 1A,B). With respect to APR-246, the susceptibility was elevated in *TP53* mutant cell lines with IC50 values of 20–30 μM, intermediate in *TP53* wild type and hemizygous cell lines with IC50 values of 50–70 μM, and reduced in *TP53* null HL-60 cells with IC50 of 80 μM. The *TP53* wild type ML-2 cells were most susceptible to APR-246 with IC50 of 22 μM (Figure 1C,D).

In order to define the most effective treatment combinations, we focused on inhibitors expected to elicit synergistic effects in combination with PTC596 based on previous studies with FLT3-, MDM2-, MCL1-, and MEK-inhibitors [20,22,23] as presented in Figure 2.

### 3.2. PTC596 Combination Treatment in AML Cell Lines

Cell viability was determined in AML cell lines treated with increasing dosages of single compounds and in combination treatments using the BMI-1 inhibitor PTC596 and a variety of targeted therapies including the *TP53* activator APR-246 (Figure 3A), the MCL1 inhibitor S63845 (Figure 3B), and the MEK inhibitor trametinib (Figure 3C). Combination indexes were calculated according to Chou Talalay (Table 2, Appendix A). Some cell lines were additionally treated with PTC596 in combination with the FLT3-ITD inhibitor PKC-412 (midostaurin), the MDM2 inhibitor HDM201, and conventional induction therapy (CI). For the *TP53* wild type cell lines, overall solid response to combination treatments was detected in the MDS-AML cell line MOLM-13, which presented moderate synergy to PTC596 in combination with S63845, trametinib, or PKC-412, as well as mild synergy to PTC596 in combination with HDM201 and APR-246. Maximal synergistic effects were present in OCI-AML3 cells treated with PTC596 and trametinib and in ML-2 cells treated with PTC596 and S63845. OCI-AML3 cells also presented moderately synergistic effects to PTC596 in combination with S63845. ML-2 response was moderately synergistic to PTC596 in combination with trametinib and mildly synergistic in combination with APR-246. Cytotoxic effects in the *TP53* wild type OCI-AML2 and the *TP53* null HL-60 cell responses were additive in the combination of PTC596 with S63845 or trametinib (Table 2).

For the *TP53* mutant cell lines, maximal synergistic effects to combination treatments were detected in SKM-1 cells with PTC596 and trametinib. Moderate synergies were present in the combinations of PTC596 and trametinib in PL-21 and MOLM-16 cells, as well as in the combination of PTC596 and S63845 in MOLM-16 and SKM-1 cells, with mildly synergistic effects in the *TP53* hemizygous PL-21 cells. PTC596 and APR-246 elicited moderate synergistic effects in PL-21 cells and mildly synergistic effects in MOLM-16 and SKM-1 cells (Table 2).

The combination of PTC596 and conventional induction therapy consisting of equimolar cytarabine and idarubicin (20–100 nM) yielded overall antagonistic effects in the tested AML cell lines (Table 2).

### 3.3. Dose-Dependent Induction of Apoptosis and Cell Death in AML Cell Lines

The effects of PTC596 and trametinib or S63845 alone and in combination on induction of apoptosis, cell cycle arrest, and cell death were determined in cytometric assays. In the *TP53* wild type cell lines strong response to combination treatments were detected in MOLM-13 cells with induction of G2 arrest by PTC596, induction of apoptosis and cell death when treated with PTC596 in combination with S63845, HDM-201, PKC-412, and trametinib (Figure 4A,B, Appendix A). OCI-AML3 cells also responded with enhanced apoptosis and cell death when treated with PTC596 in combination with S63845 or trametinib (Figure 4C,D). OCI-AML3 cells showed minor induction of G2 arrest in the cytometric assays, however, the BMI1 target gene *CDKN2A* expression was strongly upregulated in the combination treatments (Figure 4E,F) indicating induction of cell cycle arrest.

In the *TP53* mutant cell lines, strong responses to combination treatments were detected in SKM-1 cells with induction of G2 arrest by PTC596, apoptosis, and cell death when treated with PTC596 in combination with S63845 and trametinib (Figure 4G,H).

Induction of apoptosis and cell death was also present in *TP53* mutant (MOLM-16 and SKM-1) and *TP53* wild type cells (ML-2 and OCI-AML3) when treated with PTC596 and APR-246 (Figure 5).

### 3.4. PTC596 Combination Treatments in Leukemic Cells In Vitro

Strong synergistic effects were elicited in the combination of PTC596 with trametinib or S63845 in the majority of the tested AML cell lines, these treatment combinations were applied to patient derived hematological cells and normal bone marrow. A total of ten primary AML, seven secondary AML post MDS (MDS-AML), one MDS, two B-ALL, and five normal bone marrow samples were subjected to single compound and combination treatments. In the combination of PTC596 and S63845 strong cytotoxic effects were detected in two AML and one MDS-AML: AML1, classified as FLT3 mutant adverse de novo AML with 90% blast count; AML6, a relapsed t-AML; MDS-AML5, a *DNMT3A*, *TET2, NPM1* mutant secondary AML. Moderate cytotoxic effects were present in AML2 to 5, all classified as *FLT3* and *NPM1* mutant de novo AML with intermediate risk, and in the two B-ALL samples. Mild cytotoxic effects were detected in the rest of the AML and MDS-AML samples, AML7 to 9, classified as AML with *NPM1* mutant or inv(16)/CBFB-MYH11 and favorable risk, and AML 10, a *TP53* mutant AML (Figure 6A,C).

In the combination treatments with PTC596 and trametinib strong cytotoxic effects were present in four MDS-AML and one B-ALL: MDS-AML 1, 3, 6 with adverse risk, the *TP53* mutant MDS-AML2, and the B-ALL2. The *FLT3* and *NPM1* mutant AML samples (AML2 to 5) and the *TP53* mutant AML10, as well as MDS-AML4, 7, and 8 showed moderate response to PTC596 and trametinib. MDS-AML5, the *DNMT3A, TET2, NPM1* mutant secondary AML, was not susceptible to trametinib. AML7 to 9, classified as AML with mutated *NPM1* or inv(16) and favorable risk showed mild cytotoxic effects (Figure 6B,D). The cytotoxic effects elicited by the combination treatments were analyzed in the hematological cell samples correlated to CD34 positivity (Figure 6E,F) and grouped into leukemic cells with CD34 > 30% and CD34 < 30%, compared to normal bone marrow. The group with elevated CD34 positivity consisted of seven MDS-AML, one MDS, three AML, and two B-ALL samples, the group with reduced CD34 positivity consisted of eight AML samples. There was a significant correlation of CD34 positivity and response to PTC596 combination treatments (Figure 6G,H).

### 3.5. Biomarkers of Response to PTC Combination Treatments in Leukemic Cells

In order to define biomarkers of response the gene expression of *BMI1* and *MN1* as well as the protein levels of MCL1 and MEK1 were determined in the mononuclear cells isolated from AML patients at diagnosis (Table 3), and correlated to the response to PTC596 combination treatments (Figure 7A–H). *BMI1* gene expression and MEK1 protein levels were positively associated to the response to PTC596 and trametinib (Figure 7B,H), while *MN1* gene expression and MCL1 protein levels were positively associated to the response to PTC596 and S63845 (Figure 7C,E). MN1 and BMI1 gene expression were positively correlated (Figure 7I). There was a robust correlation of *MN1* gene expression with C34 positivity (Figure 7J). There was also a positive association of *BMI1* gene expression and MEK1 protein levels with CD34 positivity (Figure 7K,L).

## 4. Discussion

In order to determine the susceptibility of AML cell lines to the BMI1 inhibitor PTC596 and the *TP53* activator APR-246, a dose escalation screening was performed. Susceptibility to PTC596 varied in the tested AML cell lines with elevated susceptibility of the *TP53* null cell line HL-60, the *TP53* wild type AML cell line OCI-AML3, as well as the MDS-AML cell line MOLM-13, and reduced susceptibility of the *TP53* mutant cell lines MOLM-16, SKM-1, and PL-21 as well as the *TP53* wild type cell line ML-2. The susceptibility of *TP53* null cells to PTC596 indicated a p53 independent mechanism of action. Susceptibility of MOLM-13 and OCI-AML3 cells had been previously described for the BMI1 inhibitors PTC-209 [25] and PTC596 [26].

Susceptibility to the p53 activator APR-246 was comparable in all AML cell lines with elevated susceptibility in the *TP53* wild type ML-2 and the *TP53* mutant MOLM-16 cells indicating a p53 independent mechanism of action. APR-246 has been described as a small molecule that restores and enhances the function of mutated or wild type p53 [17,27]. APR-246 triggers an upregulation of genes involved in cell cycle control and apoptosis in *TP53* wild type and *TP53* mutant cancer cells. Moreover, APR-246 elicits p53-independent effects including production of ROS, thereby inducing oxidative stress, as well as upregulation of heat shock proteins and UPR response genes, thereby inducing ER stress [17].

In order to define the most effective treatment combinations, we focused on inhibitors expected to elicit synergistic cytotoxic effects in combination with PTC596 based on previous studies with FLT3-, MDM2-, MCL1-, and MEK- inhibitors [20,22,23]. In the current study, we found synergistic cytotoxic effects for combinations of PTC596 with APR-246, S63845, trametinib, in many AML cell lines, including the *TP53* double mutant MOLM-16, which turned out to be resistant to various other targeted therapies in the past. Here for the first time, MOLM-16 cells were susceptible to a targeted treatment with PTC596, and the effects were enhanced in combination with S63845 or trametinib.

Synergistic effects of PTC596 with S63845 and trametinib were to be expected, as PTC596 reduces *MCL-1* expression in AML cells and may influence expression of MCL1 inducers including MEK, ERK, AKT, STAT3, and STAT5 [26], while S63845 inhibits MCL1 directly, and trametinib inhibits MEK. In a previous study, we found that susceptibility to the *MCL-1* inhibitor S63845 and the MEK inhibitor trametinib correlated to the cellular MCL1 and MEK1 protein levels [20]. AML cell lines MOLM-13 and OCI-AML3, which were susceptible to S63845, had elevated MCL1 levels, while OCI-AML2 and MOLM-16, which were resistant to S63845, had reduced MCL1 levels. AML cell lines MOLM-13 and OCI-AML3, which were susceptible to trametinib, had elevated MEK levels, while OCI-AML2 and MOLM-16, both unsusceptible to trametinib, had very low MEK levels.

In the current study, we found antagonistic effects in the combination of PTC596 and conventional induction therapy in various AML cell lines. Similar antagonistic effects of PTC596 and doxorubicin or cytarabine had been previously described in mantle cell lymphoma [28]. The molecular mechanism underlying this antagonism is unclear. However, it seems that BMI1 inhibitors should not be applied in combination with doxorubicin or cytarabine, and by inference, the combination of PTC596 with other anthracyclines or nucleoside derivatives may also be unfavorable.

The effects of PTC596 and trametinib or S63845 alone and in combination on induction of apoptosis, cell cycle arrest, and cell death were determined in cytometric assays. PTC596 induced G2 arrest and apoptosis, as published previously [26]. Effects on cell cycle arrest, apoptosis, and cell death induced by PTC596 were enhanced by APR-246, S63845, trametinib, PKC412, or HDM201, as expected from the synergistic effects in the cytotoxicity assays.

Susceptibility of patient samples for PTC596 in combination with S63845 or trametinib was significant for the majority of adverse risk AML and MDS-AML samples, with least efficacy in favorable risk AML. In previous studies, BMI-1 protein levels were described to be significantly higher in patients with unfavorable cytogenetics compared with those with intermediate or favorable cytogenetics [25], and CD34 positive AML cells were more susceptible to PTC596 than mature AML cells [26]. The novel BMI-1 inhibitor PTC596 downregulated *MCL-1* and induced mitochondrial apoptosis in a p53-independent manner. PTC596 effectively killed CD34 positive AML stem/progenitor cells while sparing normal hematopoietic stem/progenitor cells. In a recent phase I study, the most frequently reported PTC596-related treatment-emergent adverse events were mild to moderate gastrointestinal symptoms, including diarrhea, nausea, vomiting, and fatigue. Only one patient treated with 10.4 mg/kg experienced dose-limiting toxicity of neutropenia and thrombocytopenia, both of which were reversible [16]. In our study, the susceptibility of leukemic cells to PTC596 in combination with S63845 or trametinib also correlated significantly to CD34 positivity of the hematological samples. As secondary AML post MDS (MDS-AML) and B-ALL are characterized by a high degree of CD34 positivity these patient subsets may profit most from a PTC596 combination treatment. There is an apparent expansion of CD34 cells during the evolution from MDS to secondary AML [29], and CD34 is a prognostic biomarker for ALL [30], hence MDS-AML and B-ALL may both be valid targets for combination treatments with PTC596 and S63845 or trametinib. In addition to CD34 positivity, *MN1* gene expression and MCL1 protein levels were biomarkers for response to PTC596 combination treatments with S63845, while *BMI1* gene expression and MEK1 protein levels were biomarkers of response to PTC596 combination treatments with trametinib. The determination of gene and protein expression levels in leukemic cells as biomarkers of response to targeted combination therapies may be helpful to optimize treatment efficacy. The combination of PTC596 and S63845 may be an effective treatment in CD34 positive AML with elevated MCL1 protein levels, while PTC596 and trametinib may be more effective in CD34 positive AML with elevated MEK1 protein levels, including *TP53* mutant AML.

## 5. Conclusions

In this study, we tested the *BMI1* inhibitor PTC-596 alone and in combination with a variety of novel targeted agents in AML cell lines and primary AML blast cells in cell viability assays. The AML cell lines included a variety of cyto-morphologic FAB subtypes as well as molecular subtypes, including *FLT3*-ITD and *FLT3* wild type, *NPM1* mutant and *NPM1* wild type, *TP53* mutant and *TP53* wild type cell lines. The results suggest that *BMI1*-inhibition by PTC-596 can induce cell cycle arrest and apoptosis in AML cells independent of *TP53* status. In fact, best response to PTC-596 was detected in *TP53* null cells, in minimally differentiated AML and in secondary AML post MDS. Induction of apoptosis and cell cycle arrest were analyzed. Effective combination treatments were validated in cytotoxicity assays with normal bone marrow and AML patient samples. Strong synergistic effects were detected in both *TP53* wild type and *TP53* mutant AML cell lines treated with PTC596 in combination with the MCL1 inhibitor S63845 or the MEK-inhibitor trametinib. In addition, mild synergistic effects were detected in both *TP53* wild type and *TP53* mutant AML cell lines treated with PTC596 and the *TP53* activator APR-246. Strong cytotoxic effects were also detected in some AML patient cells treated with PTC596 in combination with S63845 or trametinib, in particular in adverse risk AML including a *TP53* mutant MDS-AML and a relapsed t-AML. The susceptibility of leukemic cells to PTC596 in combination with S63845 or trametinib correlated significantly to CD34 positivity of the hematological samples. These results indicate that the combination of PTC596 and S63845 or trametinib may be both effective and specific treatments to target adverse risk AML, especially with elevated CD34 positivity, thus providing the rationale for initiating clinical studies evaluating these treatment combinations.

## Figures and Tables

**Figure 1 cancers-13-00581-f001:**
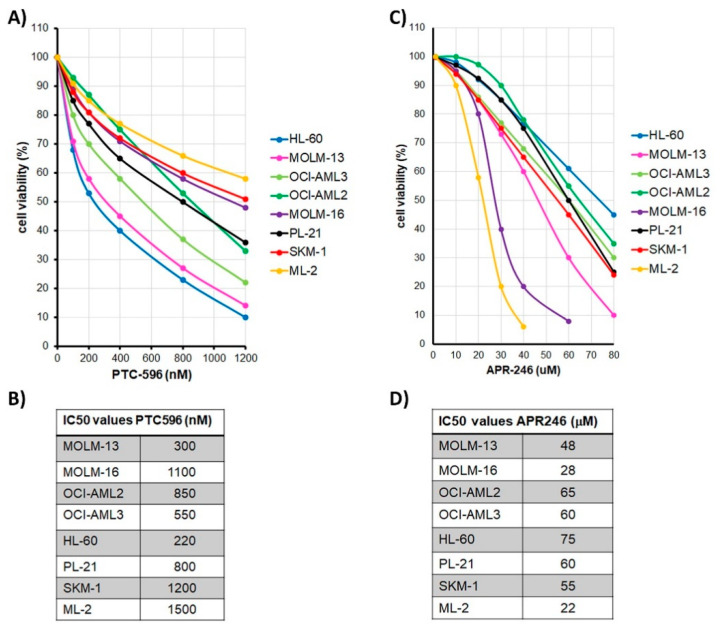
Dose response of AML cell lines treated with PTC596 or APR-246. Dose response curves and IC50 values in AML cell lines treated with PTC596 (**A**,**B**) and APR-246 (**C**,**D**). Cell viability data are average values of multiple repeat measurements per dosage. Standard deviation was 3–6%.

**Figure 2 cancers-13-00581-f002:**
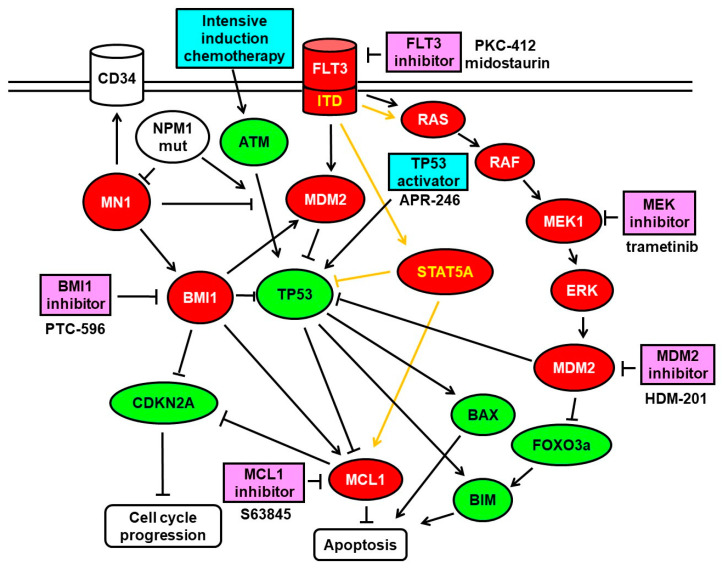
Schematic representation of the FLT3-ITD signaling pathways and downstream effects. FLT3-ITD is a constitutively active growth factor receptor signaling via PI3K-AKT, via RAS-MEK-ERK, and via STAT5 leading to cell growth and proliferation via p53 inhibition and MCL1 induction. p53 function can be reactivated by APR-246 treatment leading to inhibition of *MCL1* gene expression. MCL1 function can be inhibited by S63845, by PTC596 via BMI1 inhibition, and by APR-246 via p53 induction. Oncogenic functions are indicated in red, tumor suppressor functions in green, chemical inhibitors in pink, chemical activators in blue.

**Figure 3 cancers-13-00581-f003:**
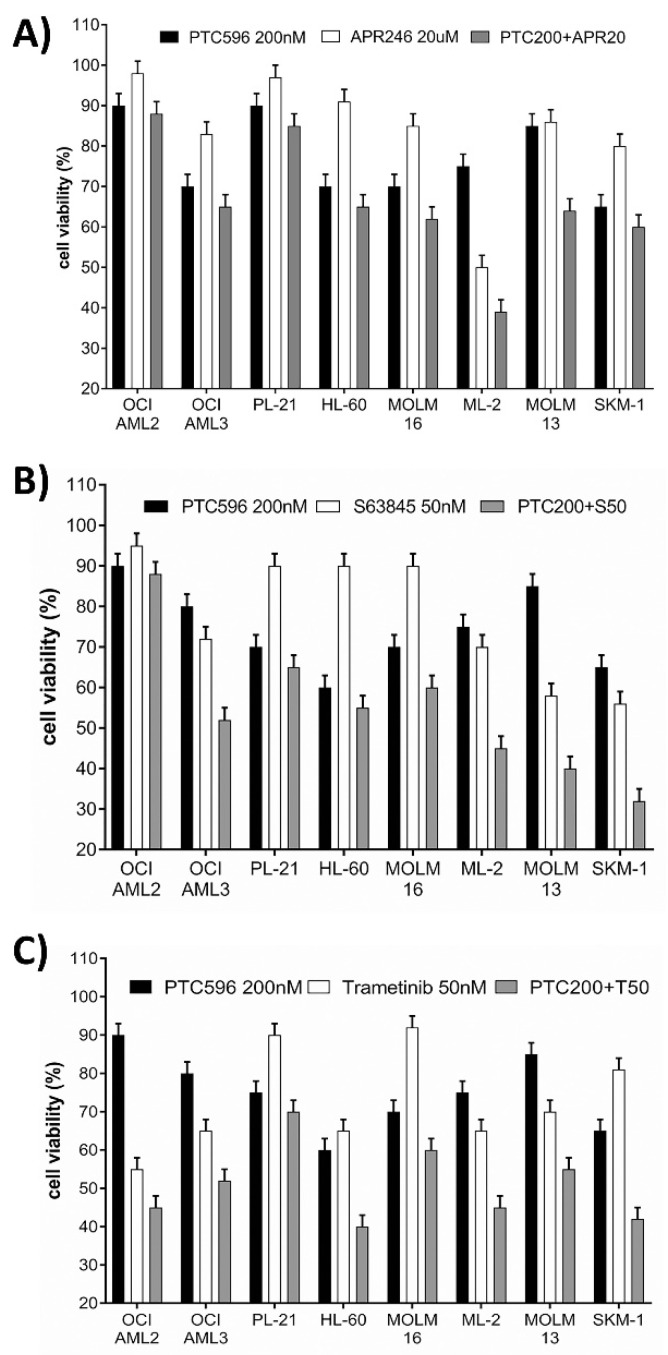
Susceptibility of AML cell lines to various treatment combinations. Cell viability was determined in AML cells after 20 h treatment with single compounds and in combination with PTC596 and APR246 (**A**), PTC596 and S63845 (**B**), and PTC596 and trametinib (**C**).

**Figure 4 cancers-13-00581-f004:**
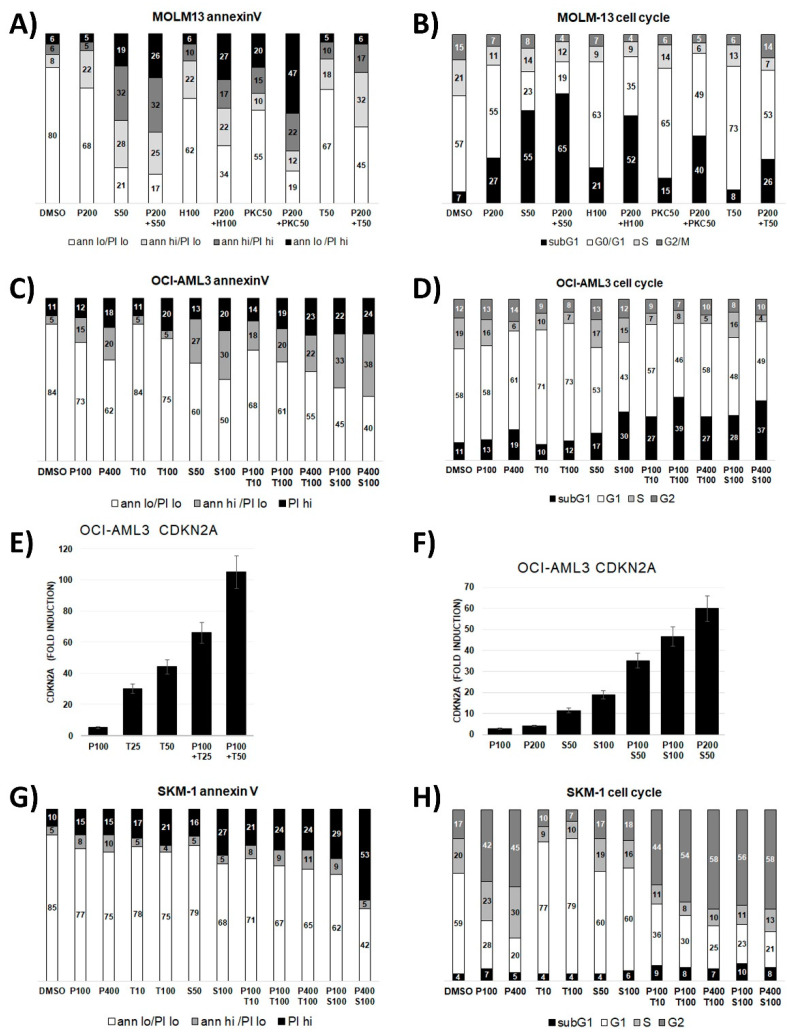
Dose-dependent induction of apoptosis and cell death in AML cells treated with PTC596 alone and in combination with targeted compounds. Cytometric analysis in AML cells treated for 20 h with PTC596 alone and in combination with PTC596 (P), S63845 (S), HDM201 (H), PKC412 (PKC), trametinib (T) to measure induction of apoptosis using annexinV and PI staining, induction of cell cycle arrest and cell death (subG1 fraction) using DAPI staining in MOLM-13 (**A**,**B**); OCI-AML3 (**C**–**F**); and SKM-1 (**G**,**H**). Relative quantitation of *CDKN2A* mRNA in OCI-AML3 cells treated for 20 h with PTC596 (P), S63845 (S), trametinib (T) alone or in combination (**E**,**F**).

**Figure 5 cancers-13-00581-f005:**
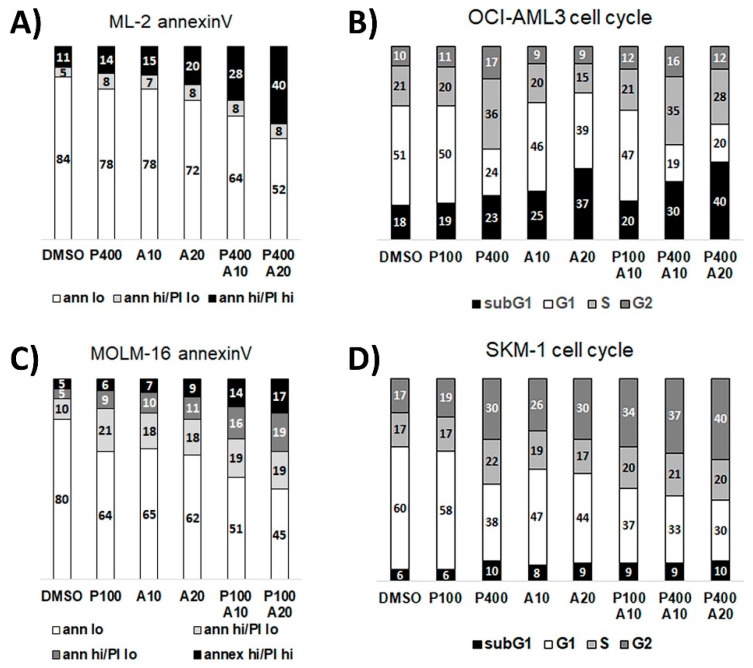
Dose-dependent induction of apoptosis and cell death in AML cells treated with PTC596 in combination with APR-246. Cytometric assays in AML cells treated for 20 h with PTC596 (P) and APR-246 (**A**) alone and in combination to measure induction of apoptosis using AnnexinV/PI staining in ML-2 (**A**) and MOLM-16 (**C**), and induction of cell death (subG1 fraction) using DAPI staining in OCI-AML3 (**B**) and SKM-1 (**D**) cells.

**Figure 6 cancers-13-00581-f006:**
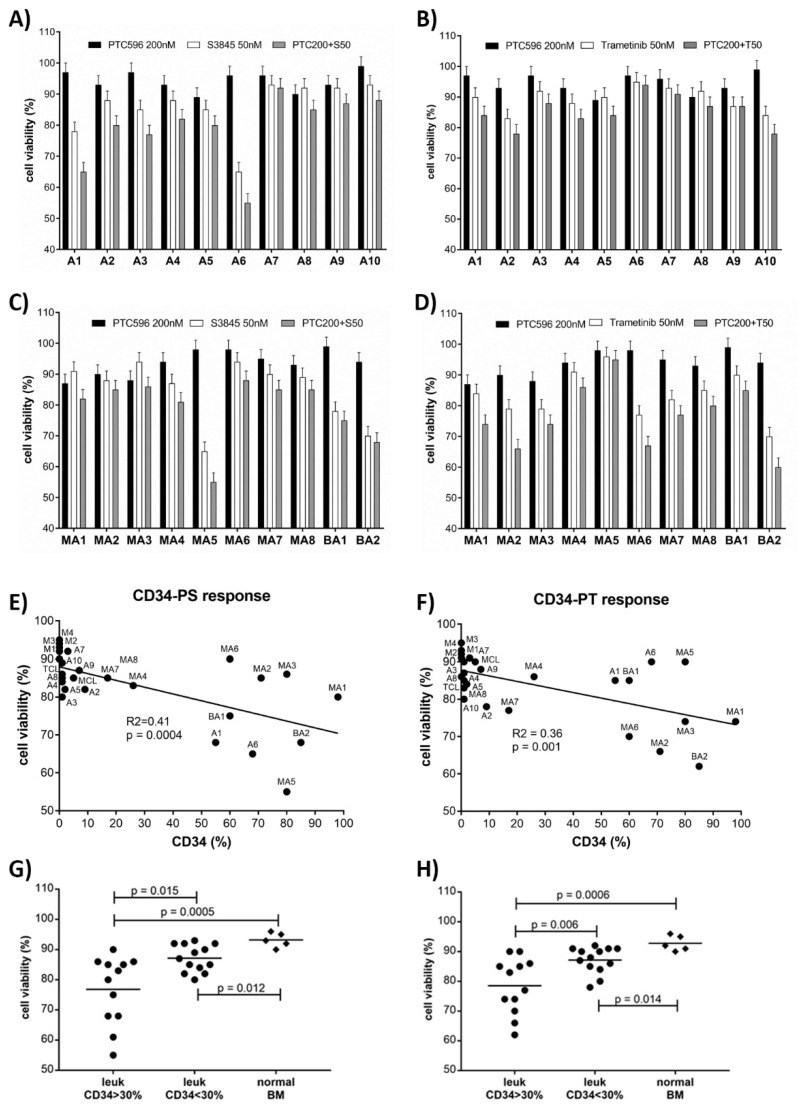
PTC596 combination treatment in patient samples in vitro. Cell viability as determined in AML, MDS-AML, and B-ALL cells treated for 20 h with single compounds and in combination with PTC596 and S63845 (**A**,**C**,**E**,**G**) or PTC596 and trametinib (**B**,**D**,**F**,**H**). Statistical significance of cell responses to combination treatment was analyzed in hematological samples correlated to CD34 positivity (**E**,**G**) and grouped into leukemic cells with CD34 > 30% or CD34 < 30% and normal bone marrow cells (**F**,**H**).

**Figure 7 cancers-13-00581-f007:**
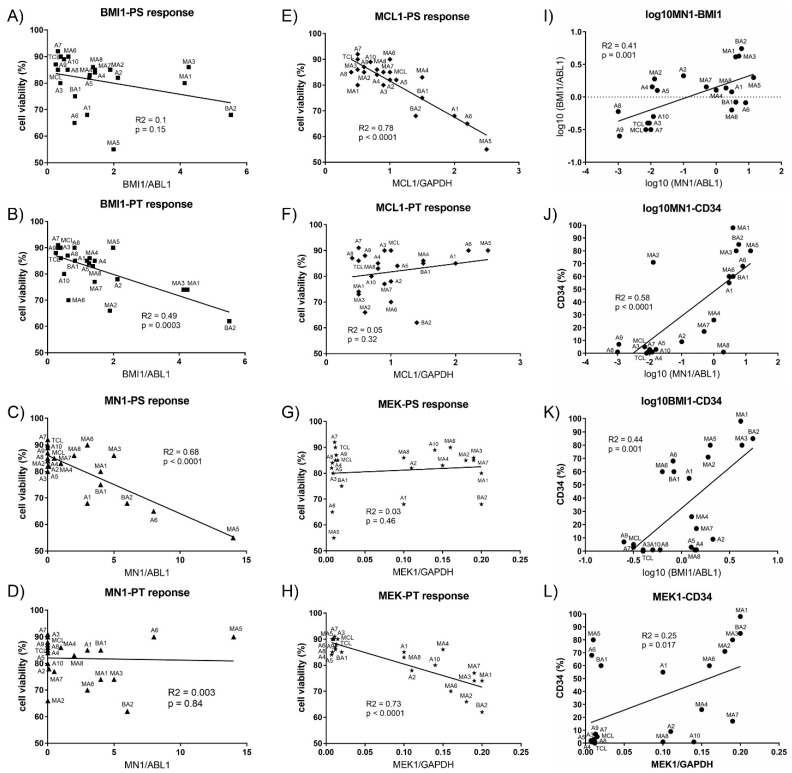
Biomarkers of response to PTC596 combination treatments. AML cell responses to combination treatments PTC596 and S63845 (**A**,**C**,**E**,**G**) or PTC596 and trametinib (**B**,**D**,**F**,**H**) were correlated to *BMI1* gene expression (**A**,**B**), *MN1* gene expression (**C**,**D**), MCL1 protein levels (**E**,**F**), and *MEK1* protein levels (**G**,**H**). Positive association of *MN1* and *BMI1* gene expression (**I**). Positive association of *MN1* gene expression (**J**), *BMI1* gene expression (**K**), and MEK1 protein levels (**L**) with CD34 positivity of leukemic cells.

**Table 1 cancers-13-00581-t001:** Genetic variants in acute myeloid leukemia (AML) cell lines.

ID	AML (FAB) Classification	Origin	*FLT3* Gene	*TP53* Gene	Other Gene Mutations	Karyotype
HL-60	M2	de novo	wt	null	*NRAS* Q61L *CDKN2A* R80X	hypotetraploid
ML-2	M4	T-ALL	wt	wt	MLL-AF6/(t6;11) *KRAS* A146T	t6;11
MOLM-13	M5a, relapse	MDS	ITD	wt	*MLL-AF9*/(t9;11)	t9;11
MOLM-16	M0, relapse	de novo	wt	V173M C238S	*MLL* V1368L *MTOR* T571K	hypotetraploid
OCI-AML2	M4	de novo	wt A680V	wt	*DNMT3A* R635W *MLL* K1751 *	+6, +8, 3q26
OCI-AML3	M4	de novo	wt	wt	*DNMT3A* R882C *NRAS* Q61L *NPM1* L287fs	+1. +5, +8
PL-21	M3	de novo	ITD P336L	wt P36fs	*KRAS* A146V	hypertetraploid
SKM-1	M5, refractory	MDS	wt	R248Q	*ASXL1*, *TET2*	del9

Abbreviations: Wild type (wt); internal tandem duplication (ITD). * is the official denotation for a stop codon.

**Table 2 cancers-13-00581-t002:** Combination index values AML cell lines (Chou and Talalay).

Cell Line	PTC596 + APR246	PTC596 + S63845	PTC596 + Trametinib	PTC596 + PKC412	PTC596 + HDM201	PTC596 + CI *
OCI-AML2	0.9–1.1	0.9–1.1	0.9–1.1	nd	0.9–1.1	nd
OCI-AML3	0.9–1.1	0.3–0.5	0.1–0.3	0.9–1.1	0.9–1.1	1.1–1.5
MOLM-13	0.9–1.1	0.4–0.6	0.3–0.5	0.3–0.5	0.7–0.9	1.1–1.5
HL-60	0.7–0.9	0.8–1.0	0.8–1.0	nd	nd	1.3–2.1
PL-21	0.4–0.6	0.7–0.9	0.4–0.6	nd	nd	>3
MOLM-16	0.7–0.9	0.5–0.7	0.3–0.5	nd	nd	nd
SKM-1	0.7–0.9	0.3–0.5	0.1–0.3	nd	nd	nd
ML-2	0.7–0.9	0.1–0.3	0.4–0.6	nd	nd	nd

Combination indexes were calculated according to Chou Talalay [24]. Interpretation of combinatorial effects: Strong synergy CI = 0.1–0.3, moderate synergy CI = 0.3–0.7, mild synergy CI = 0.7–0.9, additive effects CI = 0.9–1.1, antagonism CI > 1.1. * Conventional induction therapy (idarubicin, cytarabine).

**Table 3 cancers-13-00581-t003:** Clinical characteristics, gene expression, and protein levels in the hematological samples.

ID	Class	Stage	Mutation Profile	Karyo-Type	Risk	Sorce	Blast Count	CD34+	BMI1	MN1	MCL1	MEK1
A1	AML M1	de novo	*FLT3*	inv(4)	adv	PB	90%	55%	1.2	3.1	2.0	0.1
A2	AML M5	de novo	*FLT3* *NPM1*	norm	inter	PB	40%	9%	2.1	0.1	1.0	0.1
A3	AML M1	de novo	*FLT3* *NPM1*	norm	inter	PB	91%	1%	0.39	0.01	0.9	0.01
A4	AML M1	de novo	*FLT3* *NPM1*	norm	inter	PB	90%	1%	1.43	0.01	0.8	0.01
A5	AML M5	de novo	*FLT3 NPM1*	norm	inter	PB	90%	1%	1.26	0.02	1.1	0.01
A6	tAML	relapse	*NPM1*	norm	adv	PB	56%	68%	0.82	7.9	2.2	0.01
A7	tAML	sec	norm	inv(16)	fav	BM	30%	3%	0.32	0.01	0.5	0.01
A8	AML M5	de novo	*NPM1*	norm	fav	PB	38%	1%	0.61	0.01	0.4	0.01
A9	AML M4	de novo	*NPM1*	norm	fav	PB	26%	7%	0.25	0.01	0.6	0.01
A10	AML M4	sec	*FLT3* *TP53-G245S*	norm	adv	PB	46%	1%	0.51	0.01	0.7	0.14
MA1	AML M0	post MDS	norm	norm	adv	PB	90%	98%	4.2	3.9	0.5	0.2
MA2	AML M4	post MDS	*TP53-N259V*	T(2;5)	adv	PB	25%	71%	1.9	0.01	0.6	0.18
MA3	AML M2	post MDS	norm	del5, del7	adv	PB	60%	80%	4.2	5.1	0.5	0.19
MA4	AML	post MDS	*CEBPA* *ASXL1 EZH2 RUNX1*	norm	adv	BM	13%	26%	1.3	1.1	1.5	0.15
MA5	AML M1	post MDS	*DNMT3A TET2* *NPM1*	norm	fav	PB	81%	80%	2.0	14	2.5	0.01
MA6	MDS	de novo	*KRAS*	del(7q)	adv	PB	19%	60%	0.63	2.9	1.1	0.16
MA7	AML	post MDS	*JAK2*	norm	inter	PB	8%	17%	1.43	0.5	0.9	0.19
MA8	AML M4	post MDS	norm	norm	inter	PB	20%	53%	1.38	1.9	0.8	0.1
TCL	TCL	CR	norm	norm	na	BM	0%	0%	0.41	0.01	0.5	0.01
MCL	MCL	de novo	norm	norm	na	BM	15%	5%	0.32	0.01	1.2	0.02
BA1	B-ALL	de novo	*BCR-ABL1*	t(9;22)	na	PB	82%	63%	0.83	3.9	1.5	0.02
BA2	B-ALL	de novo	*MLL-AF4*	t(4;11)	na	BM	51%	85%	5.52	5.8	1.4	0.2
M 1–4	MM	CR	norm	norm	na	BM	0%	1%	na	na	na	na

Abbreviations: Classification (class), acute myeloid leukemia (AML), MDS-AML (M-AML), myelodysplastic syndrome (MDS), T-cell lymphoma (TCL), mantle cell lymphoma (MCL), B-cell acute lymphoblastic leukemia (B-ALL), multiple myeloma (MM), secondary (sec), adverse (adv), intermediate (inter), favorable (fav), peripheral blood (PB), bone marrow (MB), complete remission (CR). Gene expression levels for BMI1 and MN1 genes, protein levels for MCL1 and MEK1.

## Data Availability

Data is contained within the article or Appendix A.

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
