# Peer review of "BMI1-Inhibitor PTC596 in Combination with MCL1 Inhibitor S63845 or MEK Inhibitor Trametinib in the Treatment of Acute Leukemia"

_cancers, 2021, doi:10.3390/cancers13030581_

Round 1

Reviewer 1 Report

Seipel et al. submit a study of targeted therapies in AML cell lines, with a focus on defining combination synergy and correlating RNA expression with results. They use a MTT assay after 20 hours of drug exposure to define responses. They calculate synergy using Chou-Talalay methodology. Correlations with RNA expression provides some insight, but the correlations seem largely driven by a few outliers and are thus limited, even if they are logical.

Overally, the study is well done. It provides characterization of chemosensitivity across a breadth of cell lines and some insight into potential biomarkers of responsiveness.

Issues –

The selection of a 20 hour assay time-point reads out immediate cell toxicity, but not delayed apoptosis due to metabolic stress or maturation. The authors should provide more discussion of why this window was selected for analysis and what might be missed with this analysis.

The authors should better describe the synergy analysis. How many dose combinations were compared to generate the data? Did they use an 8x8 matrix or fewer combinations? Was it done in replicate?

The authors have a large amount of data to present. The figures are not always intuitive as they condense many datapoints into individual panels, and the fonts are often very small and difficult to read. In particulate, Figure 7 is one of the more interesting correlations, but it takes real effort to get your head around what is happening.  

Author Response

Reviewer 1  cancers-1061069

We appreciate the helpful comments of this reviewer.

Seipel et al. submit a study of targeted therapies in AML cell lines, with a focus on defining combination synergy and correlating RNA expression with results. They use a MTT assay after 20 hours of drug exposure to define responses. They calculate synergy using Chou-Talalay methodology. Correlations with RNA expression provides some insight, but the correlations seem largely driven by a few outliers and are thus limited, even if they are logical.

Overall, the study is well done. It provides characterization of chemosensitivity across a breadth of cell lines and some insight into potential biomarkers of responsiveness.

Revision:

  1. The selection of a 20 hours assay time-point reads out immediate cell toxicity, but not delayed apoptosis due to metabolic stress or maturation. The authors should provide more discussion of why this window was selected for analysis and what might be missed with this analysis.

 Response: This is an interesting point that has never been raised by any reviewer to any of our previous publications testing the activity, specificity and synergy of combinations of targeted compounds in leukemic cells where we always used the 20 hrs time point because it was optimal for the calculation of combination indexes. We have added the following sentence in the materials and methods section on line 136 to 139: Cell viability was determined after 20 hours of treatment using the MTT-based cell proliferation kit I (Roche). This time point was selected because the cellular responses were effectual for the calculation of combination indexes after 20 hours of treatment with two compounds in leukemic cells.

  1. The authors should better describe the synergy analysis. How many dose combinations were compared to generate the data? Did they use an 8x8 matrix or fewer combinations? Was it done in replicate?

Response: We have added the following sentence in the materials and methods section on line 139-142: For AML cell lines, four independent assays (biological replicates) with four measurements (technical replicates) per dosage were performed. For hematological patient samples two independent assays with three technical replicates per dosage were performed. For the calculation of combination indexes three dosages of PTC596 and two dosages of the other compounds were applied alone and in combination. We have added Figure S1 showing calculation of combination indexes in MOLM-13 cells.

  1. The authors have a large amount of data to present. The figures are not always intuitive as they condense many data points into individual panels, and the fonts are often very small and difficult to read. In particular, Figure 7 is one of the more interesting correlations, but it takes real effort to get your head around what is happening.  

Response: We have prepared an improved Figure 7.

Submission Date, 18 December 2020; Date of this review, January 12, 2021; Date of review report: 21 January 2021

Reviewer 2 Report

Seipel et al report in vitro cell sensitivity results to different original drugs. The tests include the evaluation of the response with MTT, Annexin V and Ki67, on 8 cell lines and 23 primary hematology samples.
Comment 1: Most of the demonstration is based on the evaluation of the response to MTT. This assay evaluates cell viability, as indicated by the authors, and also mitochondrial metabolic activity, which is not explored in the demonstration. This point seems all the more important to raise since Bmi1 has a role in mitochondrial metabolism and HL60 (the cell line most sensitive to PTC596) has a very strong mitochondrial activity. The authors should explore the metabolic impact of PTC596 and comment on their results (comparison MTT versus Annexin V for example).
Comment 2: The material and method should include more details to allow the experiments to be replicated. For example, details are missing on molecular biology techniques for the study of patient mutations and on the targets studied (exon from TP53?), information on cell culture methods (temperature, 02), the time of cell treatment by the different molecules, the cell density for the different techniques?
Comment 3: the quality of the figures needs to be improved enormously :
-Figure 1 is of poor quality;
-figure 2 is difficult to read;
the legend of the Y-axis of figure 3A is missing as well as an explanation of the statistical tests used (P+A vs. which group?).
-in figures 4 and 5, I did not find the number of replicates per experiment nor statistical tests. In fact, the synergistic effects are difficult to appreciate and visually seem more additive.
-Figure 7 is uninterpretable. This defect is strongly penalizing, so much so that I could not evaluate the part of the work on biomarkers (when it should be very interesting).

Author Response

Reviewer 2 cancers-1061069

We appreciate the helpful comments of this reviewer.

Seipel et al report in vitro cell sensitivity results to different original drugs. The tests include the evaluation of the response with MTT and Annexin V, on 8 cell lines and 23 primary hematology samples.

Revision:
Comment 1: Most of the demonstration is based on the evaluation of the response to MTT. This assay evaluates cell viability, as indicated by the authors, and also mitochondrial metabolic activity, which is not explored in the demonstration. This point seems all the more important to raise since Bmi1 has a role in mitochondrial metabolism and HL60 (the cell line most sensitive to PTC596) has a very strong mitochondrial activity. The authors should explore the metabolic impact of PTC596 and comment on their results (comparison MTT versus Annexin V for example).

Response: Insight into the indispensable role of BMI1 in normal cells has come from studies of BMI KO mice that displayed a progressive decrease in the number of hematopoietic cells and neurologic abnormalities. BMI1, a predominantly nuclear protein, regulates cellular processes critical for cell growth, cell fate decision, development, senescence, aging, DNA damage repair, apoptosis, and self-renewal of stem cells. In addition, an extranuclear pool of BMI1 localizes to the inner mitochondrial membrane and directly regulates mitochondrial RNA (mtRNA) homeostasis and bioenergetics.

A putative correlation of mitochondrial activity and BMI1 sensitivity is intriguing. If this correlation applies to AML, ML-2 cells should have reduced mitochondrial activity. It would be interesting to explore this possibility. However, this lies outside the scope of this project, and is not feasible within this revision.

For the biomarkers of response we focused on correlations present in the AML patient samples. We found correlations of response with BMI1 and MN1 gene expression, as well as MCL1 and MEK1 protein levels. We concluded that the combination treatment with the BMI1 inhibitor PTC596 and the MCL1 inhibitor S63845 may be an effective treatment in CD34+ adverse risk AML with elevated MN1 gene expression and MCL1 protein levels, while combination treatment with BMI1 inhibitor PTC596 and the MEK inhibitor trametinib may be more effective in CD34+ adverse risk AML with elevated BMI1 gene expression and MEK protein levels.

Comment 2: The material and method should include more details to allow the experiments to be replicated. For example, details are missing on molecular biology techniques for the study of patient mutations and on the targets studied (exon from TP53?), information on cell culture methods (temperature, 02), the time of cell treatment by the different molecules, the cell density for the different techniques?

Response: We have added the following sentence to the material and methods section, line 117/118: The AML cells were analyzed at the central hematology laboratory of the University Hospital Bern according to state of the art techniques (Shumilov et al., 2018). Mutational screening for FLT3, NPM1, TP53 and conventional karyotype analysis of at least 20 metaphases were performed in all samples. More recent samples were analyzed by NGS sequencing of the myeloid panel genes. We have added the specific TP53 mutations in table 4. The time of treatment was 20 hours in all assays. This information has been added to the figure legends. Additional information on the cell culture techniques has been added in the materials and methods section, line 128.

Comment 3: the quality of the figures needs to be improved enormously :
-Figure 1 is of poor quality;
-figure 2 is difficult to read;
the legend of the Y-axis of figure 3A is missing as well as an explanation of the statistical tests used (P+A vs. which group?).
-in figures 4 and 5, I did not find the number of replicates per experiment nor statistical tests. In fact, the synergistic effects are difficult to appreciate and visually seem more additive.
-Figure 7 is uninterpretable. This defect is strongly penalizing, so much so that I could not evaluate the part of the work on biomarkers (when it should be very interesting).

Response: All Figures have been improved. We have removed the ttest values in Figure 3, as the combination indexes shown in table 2 are more appropriate to indicate synergistic effects for the cell lines.

Submission Date, 18 December 2020; Date of this review, January 12, 2021; Date of review report: 21 January 2021

Reviewer 3 Report

In this manuscript, Seipel and colleagues tested the anti-leukemia efficacy of BMI1 inhibitor PTC596 in combination with other targeted therapies using AML cell lines and patient samples. They identified the combination of PTC596 with MCL1 inhibitor S63845 or MEK inhibitor trametinib may be an effective treatment in CD34+ adverse risk AML with specific biomarkers. The manuscript is well presented. However, although the study is interesting, the authors should address the following concerns before the study can be considerate for publication.

  1. It is interesting to see the effect of PTC596 combination in CD34+ leukemic cells but not normal bone marrow cells (Figures 6G-H). It is necessary to include purified CD34+ cells from normal bone marrow or cord blood as a control.
  2. More detailed information about the experimental design should be indicated either in Method or Figure Legend.
  3. If the patient derived cells are cultured before treatment or analysis?
  4. For how long and in what density were the AML cell lines and patient derived cells treated for different experiments?
  5. In Figure 3, please describe the details of ttest and how you define ns, *, **, and ***.
  6. In Figures 4-5, all the abbreviations (T, S, numbers…) should be clearly defined in figure legend.
  7. Method for protein levels measurement needs to be included.
  8. Gating strategies of AnnexinV/PI staining and cell cycle need to be shown.
  9. Figure 7 is hard to read. Please provide a figure with a higher resolution.

Author Response

Reviewer 3 cancers 1061069

We appreciate the helpful comments of this reviewer.

In this manuscript, Seipel and colleagues tested the anti-leukemia efficacy of BMI1 inhibitor PTC596 in combination with other targeted therapies using AML cell lines and patient samples. They identified the combination of PTC596 with MCL1 inhibitor S63845 or MEK inhibitor trametinib may be an effective treatment in CD34+ adverse risk AML with specific biomarkers. The manuscript is well presented. However, although the study is interesting, the authors should address the following concerns before the study can be considerate for publication.

Revision:

  1. It is interesting to see the effect of PTC596 combination in CD34+ leukemic cells but not normal bone marrow cells (Figures 6G-H). It is necessary to include purified CD34+ cells from normal bone marrow or cord blood as a control.

Response: We agree that it would be interesting to test CD34+ cells from normal bone marrow. However, this will not be feasible in this revision time frame of 10 days. However, to address this point, we added a sentence in the discussion on line 377 to 381 in regard to the PTC596 phase I study in solid tumors (Shapiro et al., 2021, in press). Overall, PTC596 was well tolerated. The most frequently reported PTC596-related treatment-emergent adverse events were mild to moderate gastrointestinal symptoms, including diarrhea (54.8%), nausea (45.2%), vomiting (35.5%), and fatigue (35.5%). Only 1 patient treated with 10.4 mg/kg experienced dose-limiting toxicity of neutropenia and thrombocytopenia, both of which were reversible.

  1. More detailed information about the experimental design should be indicated either in Method or Figure Legend.

Response: We have extended the descriptions in the Materials and Methods section on and included more detailed information in the Figure legend.

  1. If the patient derived cells are cultured before treatment or analysis?

Response: We have added a sentence in the Materials and Methods section, line 131/132: For assays with patient derived mononuclear cells, the cells were cultured for 20hrs prior to treatment.

  1. For how long and in what density were the AML cell lines and patient derived cells treated for different experiments?

Response: The time of treatment was 20 hours in all assays. This information has been added to the figure legends. Additional information on the cell culture techniques has been added in the materials and methods section.

  1. In Figure 3, please describe the details of ttest and how you define ns, *, **, and ***.

Response: We have removed the ttest values in Figure 3, as the combination indexes shown in table 2 are more appropriate to indicate synergistic effects for the cell lines.

  1. In Figures 4-5, all the abbreviations (T, S, numbers…) should be clearly defined in figure legend.

Response: Abbreviations have been defined in the figure legends.

  1. Method for protein levels measurement needs to be included.

Response: Measurement of protein levels can be found in the materials and methods section, line 159, chapter 2.5. Enzyme-Linked Immunosorbent Assay (ELISA). 

  1. Gating strategies of AnnexinV/PI staining and cell cycle need to be shown.

Response: We have added the imager files for the MOLM-13 cells in the supplement Figures 2 and 3.

  1. Figure 7 is hard to read. Please provide a figure with a higher resolution.

Response: Figure 7 has been improved. 

Submission Date, 18 December 2020; Date of this review, January 12, 2021; Date of review report: 21 January 2021

Round 2

Reviewer 3 Report

My concerns have been addressed.